# Greater nurse density correlates to higher level of population ageing globally, but is more prominent in developed countries

Wenpeng You[1,2,3]*, Frank Donnelly[1]

**1** Adelaide Nursing School, the University of Adelaide, Adelaide, Australia, **2** Heart and Lung, Royal Adelaide Hospital, Adelaide, Australia, **3** Adelaide Medical School, the University of Adelaide, Adelaide, Australia

* wenpeng.you@adelaide.edu.au

## Abstract

### Background

Representing over 50% of the healthcare workforce, nurses provide care to people at all ages. This study advances, at a population level, that high levels of nursing services, measured by nurse density may significantly promote population ageing measured by the percentage of a population over 65 years of age (65yo%).

### Methods

Population level data was examined to explore the correlation between nurse density and 65yo%. The confounding impacts on ageing such as the effects of economic affluence, physician density, fertility rate, obesity and urban advantages were also considered. Scatter plots, bivariate correlation, partial correlation and multiple linear regression analyses were performed for examining the correlations.

### Results

Nurse density correlated to 65yo%; this relationship was independent of other influences such as fertility rate, economic affluence, obesity prevalence, physician density and urban advantages. Second to fertility rate, nursing density had the greatest influence on 65yo%. The predicting and confounding variables explain 74.4% of the total 65yo% variance. The universal correlations identified in country groupings suggest that low nurse density may be a significant global concern.

### Conclusions

While nurse density might contribute significantly to 65yo% globally, the effect was more prominent in developed countries. Ironically, countries with higher nurse densities and therefore greater levels of 65yo%, were countries with an increased need for more nursing staff. To highlight the profound implications for the role the nursing profession plays especially at a time of global nursing shortage, further study into the effects of long-run elasticity of nurse staffing level on population ageing may be needed. For instance, what percentage

**Data Availability Statement:** All relevant data sources are described within the manuscript.

**Funding:** The author(s) received no specific funding for this work.

**Competing interests:** The authors have declared that no competing interests exist.

of nursing staff increase would be required to meet every 1% increase of an ageing population.

## Introduction

Over the last 200 years, people around the world have achieved impressive progress in health represented by an increase of the number of people aged 65 and over. Population ageing refers to changes in the age composition of a population which increases the proportion of older persons. It has become the 21st century's dominant demographic measure. Demographically, the most often used index of population ageing is the percentage of population segment aged 65 and over (65yo% hereafter) [1, 2].

It is well-established that declining fertility rates and rising life expectancy are the primary causes for population ageing [3–6]. Without sustainable numbers of younger people to replace the ageing population, the relative size of the population segment aged 65 years old or over will continue to increase. Improved economic circumstances have been exacerbating this impact because of increased welfare and life satisfaction [7]. Other studies identify factors, such as education [8] and genetic traits [9] that may also be contributing factors impacting population ageing.

The evolution of the nursing workforce into an independent health care profession is evident as nurses now account for over 50% of the global healthcare industry [10]. Worldwide, nurses have and will continue to play a critical role in health promotion, disease prevention and delivery of a comprehensive range of healthcare services [10]. Within primary, secondary and tertiary healthcare settings, the scope of nursing healthcare has afforded nurses' opportunities to lead and coordinate patient care provided by multidisciplinary health professionals. Therefore, the quality of healthcare from nurses should be considered a core indicator of a populations level of healthcare. Previous studies, for example, have revealed that nursing workforce plays a significant role in reducing infant and neonatal mortalities [11] and life expectancy at birth and at 65 years old [12, 13]. An outcome from the role of nursing workforce in promoting population health has been an increase in the number of who survive past 65 years old.

Ironically, as the population of many countries pushing past 65yo increase, these countries have the burdens of an ageing population started to further impact available healthcare systems. In recent years the level of demand has grown, at a much faster pace [14]. This impact has contributed to the global nursing staff shortage crisis because old people have higher healthcare dependency on healthcare professionals, especially nursing staff. Studies in individual healthcare facility settings or at community levels have reported that nursing shortages are a chronic issue [15], and this was highlighted during the COVID-19 pandemic period [16].

The challenge for health authorities is to take strategic actions to provide sufficient healthcare to an older population with high demand of nursing care. This study helps to illustrate and quantify the role of nursing workforce in increasing the proportion of people over 65 years old. In this study, we hypothesized that, globally, nursing healthcare services may be a significant contributor for increasing the portion of people to live over 65 years. We tested this hypothesis by examining the statistical relationship between nurse density and the 65yo%, globally and regionally. With reference to previous studies into healthcare outcomes of nursing staff, economic affluence, obesity prevalence, physician density, total fertility rate and urbanization were incorporated as potential confounding factors.

## Materials and methods

### Data sources

The country specific data published by the agencies of the United Nations (UN) were extracted for this ecological study. Country in this study does not necessarily indicate political independence but only refers to the geographic territory or region which reported data on health, demography and economic situation to the World Bank [17]. Except for the obesity prevalence rate which was extracted from the World Health Organization [18], all other variables (nurse density, population ageing level, GDP PPP, physician density, total fertility rate and urbanization) were downloaded from the World Bank database [19].

1. The dependent variable, population ageing level indexed with the percentage of population aged 65 and above (65yo%) in 2020 [20].

    The DataBank of the World Bank is an analysis and visualisation tool that contains collections of time series data on a variety of topics, including socioeconomics, key health, nutrition and demography statistics. Population is based on the de facto definition of population, which counts all residents regardless of legal residency status or citizenship.

2. The independent variable, nurse density measured with the number of nurses and midwives per 1,000 people [21].

    To reduce random errors at the time of data collection, we averaged the number of nurses and midwives for the period between 2014 and 2018 in each country for our data analyses.
    Empirically and as per recent studies, economic affluence, obesity, physician service access, total fertility rate and urban living have been either positive or negatively associated with people's health affecting their opportunities surviving 65 years old. Therefore, they were included as the potential confounders while we analysed the independent role of nursing workforce in promoting population ageing:

3. Economic affluence, expressed with the gross domestic product (GDP) per capita [22].

    The economic affluence is specifically indexed as GDP PPP per capita (current international $ for purchasing power parity) in 2014. GDP PPP is included as the index of economic affluence in this study as it has the advantage over GDP per capita because it considers the relative cost of local goods, services and inflation rates of the country. Therefore, GDP PPP is more associated with the life quality and wellbeing of individuals who live in different countries [7, 23, 24].

4. Obesity prevalence rate, measured with the percentage of adult individuals with the body mass index (BMI) equal to or exceeding 30 kg/m$^2$ in 2014 [18].

    Obesity is a result of metabolic imbalances which increases the risk for obese individuals to develop medical complications and die younger than 65 years old [25].

5. Physician density, expressed with the number of physicians per 1,000 people [26].

    Physician healthcare access determines not only the level of primary healthcare services, but also the diagnosis and treatment of patients' health conditions at the secondary and tertiary levels. Therefore, physician and nursing healthcare services may be seen to confound with each other for maintaining and improving people's health to facilitate them to survive 65 years old [27].

6. Fertility rate (total), representing the total number of children born to a woman [28].

    It is reasonable to note that falling fertility rates are one of the major determinants of population aging [3–6]. Low fertility rates result in smaller youth cohorts, which creates an

imbalance in the demographic structure: older age groups expand and become more populous than their younger counterparts [29].

7. Urbanization, indexed with the percentage of total population living in urban areas in 2014 [30].

Urbanization represents a major demographic shift which is characterised by lifestyle changes, for instance lack of physical activity and poor diet patterns [31–33]. Studies controversially showed that people living in both metropolitan areas and rural areas have greater life expectancy and higher percentage to survive 65 years old. However, urban residents experience relatively larger gains in life expectancy than those in rural areas [34].

The relevant United Nations agencies offer free online access to data required for the analyses such as in this study. The data can only be identifiable to the level of population. They are not identifiable or re-identifiable to the individual participant, their family or community. Therefore, there is no need to obtain ethical approval or consent during the entire study process.

## Data selection

We extracted the country specific data on independent variable (nurse density), dependent variable (population ageing level) and 5 potential confounders (GDP PPP, obesity prevalence, physician density, total fertility rate and urbanization). For each variable, we extracted the data from all the countries where data were available from the websites of the United Nation agencies.

Following the country list created by the World Bank, we aligned all the 7 variables from the individual countries and obtained a full set of data comprising 215 countries. Each country was treated as an individual study subject in all the data analysis models. However, not all the countries (subjects) have all the information for all the variables. The numbers of countries/ subjects (sample size) included for analysing the correlations to other variables may differ as such.

## Data analysis

To assess the correlation between the nurse density and the 65yo%, the analysis proceeded in the following 5 models [27, 35–38]:

1) Scatter plots were explored with the raw data in Microsoft Excel$^®$ for examining and visualizing the strength, shape and direction of correlation of nurse density to 65yo%. Additionally, the scatter plots graph allowed an examination of data quality, for example, if there are any unexpected gaps in the data and if there are any extreme outlier points.
   To reduce the skewness of our original data for valid statistical analysis results produced in the following data analysis models, all the 7 variables were log-transformed.

2. Bivariate (Pearson's r and nonparametric) correlations were conducted to examine the directions and strengths of the correlations between all the variables.

3. Partial correlation of Pearson's moment-product approach was performed to identify the independent correlation relationships between variables. We alternated each of the 6 variables (nurse density, GDP PPP, obesity, physician density, total fertility rate and urbanization) as the independent predictor when all the other 5 variables are included as the potential confounding factors. And then, we alternated each individual variable as the controlled variable to assess the relationship between 65yo% and each of the 5 variables.

4. Standard multiple linear regression (enter) was performed to describe the correlations between the dependent variable (65yo%) and the predicting variables. To explore if and how much nurse density could statistically explain the individual relationships between nurse density and GDP PPP, obesity, physician density, total fertility rate and urbanization, the enter multiple linear regression was performed to calculate the correlations between nurse density and the confounding variables when nurse density is "added" and "not added" as a predicting variable respectively.

Subsequently, standard multiple linear regression (stepwise) is performed to select the most significant predicting variable(s) for 65yo% when nurse density was "added" and "not added" as a predicting variable respectively.

5. The universal correlations between nurse density and 65yo% were explored and compared in different country groupings:

1) the World Bank income classifications: high income, upper middle income, low-middle income and low income;

2) the UN common practice on defining the developed and developing countries [39]; Fisher's r-to-z transformation was applied to compare the nurse density- 65yo% correlations in the developed countries and the developing countries.

3) the WHO regional classifications: Africa (AFR), Americas (AMR), Eastern Mediterranean (EMR), Europe (EU), South-East Asia (SEAR) and Western Pacific (WPR) [40];

4) countries with the strong contrast in terms of geographic distributions, per capita GDP levels and/or cultural backgrounds. We analysed the correlation in the 6 country groupings: Asia Cooperation Dialogue (ACD) [41]; the Asia-Pacific Economic Cooperation (APEC); the Arab World [42], European Economic Area (EEA) [43], countries with English as the official language (government websites), European Union (EU) [44], Latin America [45], Latin America and the Caribbean (LAC) [45], the Organisation for Economic Co-operation and Development (OECD) and non-OECD group [46].

Bivariate correlations, partial correlation, multiple linear regression analyses (enter and stepwise) were conducted with SPSS v. 28. The significance is kept at the 0.05 level, but 0.01 and 0.001 levels ae also reported. Standard multiple linear regression analysis criteria are set at probability of F to enter $\leq$ 0.05 and probability of F to remove $\geq$ 0.10.

## Results

The relationship identified in the scatterplots between nurse density and population ageing was a strong correlation ($R^2$ = 0.5268 (r = 0.7258), p<0.001, n = 177, Fig 1). The nursing workforce explains 52.68% of population ageing variance.

This strong relationship between nurse density and population ageing identified in the scatterplots was confirmed by the subsequent bivariate correlation analyses with the log-transformed data.

Globally, nurse density significantly correlated to 65yo% (r = 0.694 and rho = 0.724, p<0.001 respectively in Pearson and non-parametric analyses, Table 1).

When predicting and confounding variables (nurse density, GDP PPP, obesity, total fertility rate, physician density and urbanization) were individually correlated to population ageing, while keeping the other 5 variables statistically constant, nurse density, GDP PPP and total fertility rate remained significant correlations to 65yo% (r = 0.194, p< 0.05; r = 0.223, p< 0.01 and r = 0.609, p < 0.001 respectively, Table 2-1). Neither obesity prevalence nor urbanization

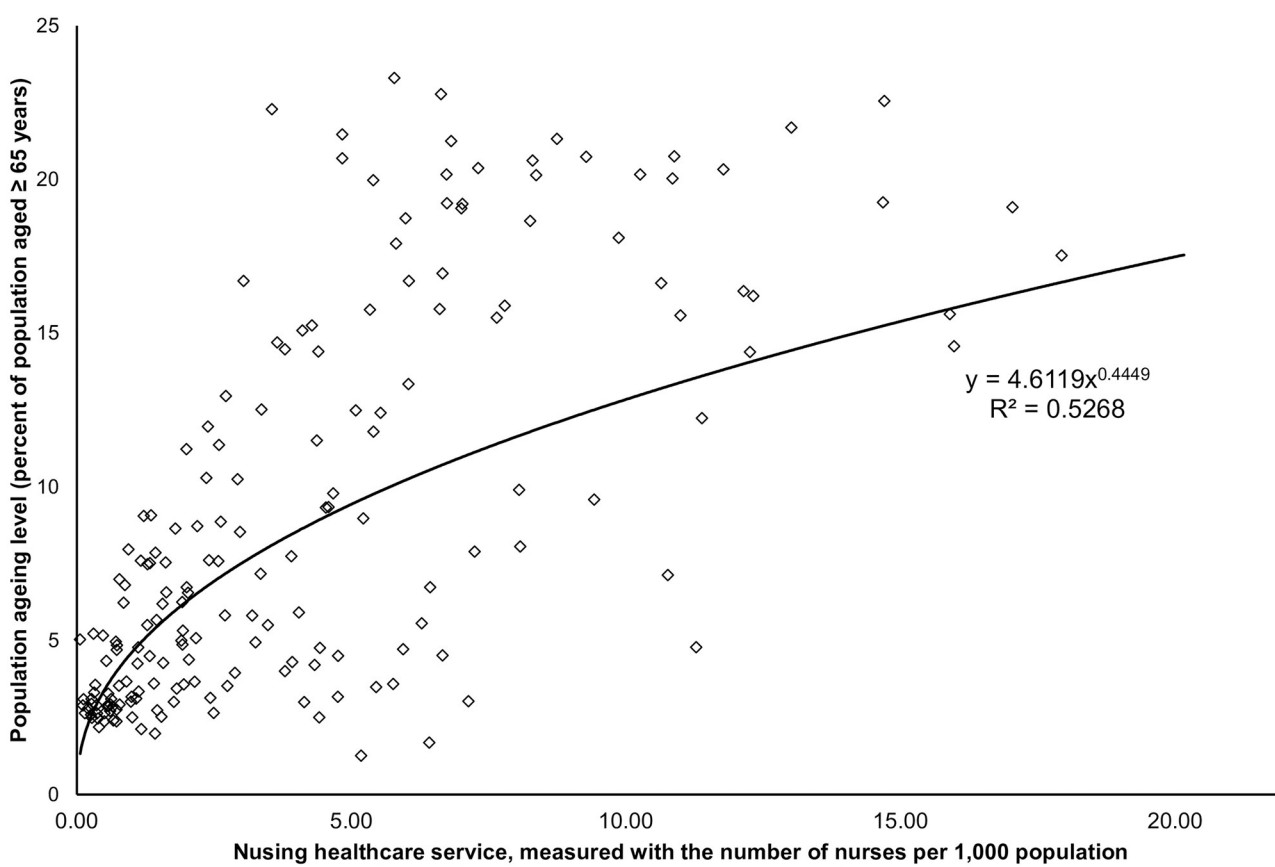

**Fig 1. The relationship between nurse density and population ageing level.**

**Table 1. Pearson (above diagonal) & non-parametric (below diagonal) correlation matrix for all variables.**

|  | Nurse density | Population ageing | GDP PPP | Obesity prevalence | Physician density | Total fertility rate | Urbanization |
|---|---|---|---|---|---|---|---|
| Nurse density | 1 | 0.694*** | 0.774*** | 0.508*** | 0.784*** | -0.735*** | 0.547*** |
| Population ageing | 0.724*** | 1 | 0.671*** | 0.365*** | 0.759*** | -0.852*** | 0.494*** |
| GDP PPP | 0.797*** | 0.686*** | 1 | 0.502*** | 0.839*** | -0.799*** | 0.720*** |
| Obesity prevalence | 0.465*** | 0.365*** | 0.483*** | 1 | 0.500*** | -0.391*** | 0.546*** |
| Physician density | 0.810*** | 0.810*** | 0.831*** | 0.445*** | 1 | -0.841*** | 0.626*** |
| Total fertility rate | -0.745*** | -0.872*** | -0.777*** | -0.352*** | -0.816*** | 1 | -0.521*** |
| Urbanization | 0.600*** | 0.506*** | 0.757*** | 0.584*** | 0.630*** | -0.522*** | 1 |

Significance level:

***$p < 0.001$; n ranges between 176 and 212.

Data source and definition: Nurse density, expressed with the number of nurses and midwives per 1,000 population (the Word Bank); Population ageing, presented with the percent of population aged 65 and above (the Word Bank); GDP PPP, the per capita purchasing power parity (PPP) value of all final goods and services produced within a territory in a given year (the Word Bank); Obesity prevalence, the percentage of population with BMI $\geq 30$ prevalence (WHO Global Health Observatory); Physician availability, the number of nurses and midwives per 1,000 population (the Word Bank); Total fertility rate, representing the number of children that are born to a woman, the World Bank; Urbanization, the percentage of population living in urban area (the Word Bank);

All the data were log-transformed for correlation analysis.

**Table 2. Partial correlation coefficients between population ageing and nurse density with individual and different combinations of controlled variables.**

| Variables | Table 2–1: Nurse density, GDP PPP, obesity prevalence, physician density, total fertility rate and urbanization were alternated as the predicting variable for calculating its relationship with population ageing level while the other 4 variables were kept statistically constant. | | | | | | | | | | | |
|---|---|---|---|---|---|---|---|---|---|---|---|---|
| | Population ageing (65yo%) | | Population ageing (65yo%) | | Population ageing (65yo%) | | Population ageing (65yo%) | | Population ageing (65yo%) | | Population ageing (65yo%) | |
| | r | p | r | p | r | p | r | p | r | p | r | p |
| Nurse density | 0.194 | <0.050 | - | - | - | - | - | - | - | - | - | - |
| GDP PPP | - | - | -0.223 | <0.010 | - | - | - | - | - | - | - | - |
| Obesity prevalence | - | - | - | - | -0.038 | 0.626 | - | - | - | - | - | - |
| Physician density | - | - | - | - | - | - | 0.119 | 0.120 | - | - | - | - |
| Total fertility rate | - | - | - | - | - | - | - | - | -0.609 | < 0.001 | - | - |
| Urbanization | - | - | - | - | - | - | - | - | - | - | 0.147 | 0.055 |
| Variables | Table 2–2: Nurse density, GDP PPP, obesity prevalence, physician density, total fertility rate and urbanization were alternated as the individual potential confounder for exploring the partial correlations between population ageing level and the other 4 independent/confounding variables. | | | | | | | | | | | |
| | Population ageing (65yo%) | | Population ageing (65yo%) | | Population ageing (65yo%) | | Population ageing (65yo%) | | Population ageing (65yo%) | | Population ageing (65yo%) | |
| | r | p | r | p | r | p | r | p | r | p | r | p |
| Nurse density | - | - | 0.371 | < 0.001 | 0.634 | < 0.001 | 0.243 | < 0.001 | 0.189 | < 0.010 | 0.582 | < 0.001 |
| GDP PPP | 0.295 | < 0.001 | - | - | 0.606 | < 0.001 | 0.097 | 0.196 | -0.031 | 0.683 | 0.523 | < 0.001 |
| Obesity prevalence | 0.020 | 0.793 | 0.043 | 0.569 | - | - | -0.027 | 0.720 | 0.064 | 0.394 | 0.130 | 0.084 |
| Physician density | 0.482 | < 0.001 | 0.487 | < 0.001 | 0.716 | < 0.001 | - | - | 0.150 | 0.043 | 0.664 | < 0.001 |
| Total fertility rate | -0.701 | < 0.001 | -0.709 | < 0.001 | -0.828 | < 0.001 | -0.607 | < 0.001 | - | - | -0.802 | < 0.001 |
| Urbanization | 0.190 | < 0.010 | 0.021 | 0.782 | 0.378 | < 0.001 | 0.036 | 0.629 | 0.111 | 0.129 | - | - |

- Controlled variable; Table 2–1 df = 169; Table 2–2 df range: 173 to 188

Data source and definition Data source and definition: Nurse density, expressed with the number of nurses and midwives per 1,000 population (the Word Bank); Population ageing, presented with the percent of population aged 65 and above (the Word Bank); GDP PPP, the per capita purchasing power parity (PPP) value of all final goods and services produced within a territory in a given year (the Word Bank); Obesity prevalence, the percentage of population with BMI ≥30 prevalence (WHO Global Health Observatory); Physician availability, the number of nurses and midwives per 1,000 population (the Word Bank); Total fertility rate, representing the number of children that are born to a woman, the World Bank; Urbanization, the percentage of population living in urban area (the Word Bank);

All the data were log-transformed for correlation analysis.

showed an independent correlation to 65yo% while the other 5 variables were kept statistically constant (Table 2-1).

Moreover, nurse density was in constant and significant correlation to 65yo% when GDP PPP, obesity prevalence, total fertility rate, physician density and urbanization were individually controlled (r = 0.371, p< 0.001; r = 0.634, p< 0.001; r = 0.243, p< 0.001; r = 0.189, p< 0.01 and r = 0.582, p< 0.001 respectively, Table 2-2).

These partial correlation analysis results suggest that, statistically, the individual, or the combined confounding effects of GDP PPP, obesity prevalence, total fertility rate, physician density and urbanization did not affect the significant relationship of nurse density to people increasing over 65yo%.

Standard multiple linear regression (enter) analysis was applied to further predict 65yo% when nurse density, GDP PPP, obesity, physician density and urbanization were considered as the predicting variables. When nurse density was "not added" as one of the predicting variables, physician density and total fertility rate were in significant correlations to 65yo% (β = 0.230, p<0.01 and β = -0.770, p<0.001 respectively). GDP PPP, obesity prevalence and urbanization showed weak and insignificant correlations to 65yo% (Table 3-1). When nurse density

**Table 3. Multiple linear regression results to show predicting effects of independent variables and identify the significant predictors of population ageing level.**

**Enter**

| | Population ageing (65yo%) | | | |
| | Nurse density (not added) | | Nurse density (added) | |
| Variable | Beta | Significance | Beta | Significance |
|---|---|---|---|---|
| Nurse density | Not added | | 0.178 | < 0.050 |
| GDP PPP | -0.105 | 0.065 | -0.239 | < 0.010 |
| Obesity prevalence | 0.029 | 0.563 | 0.008 | 0.874 |
| Physician density | 0.230 | < 0.010 | 0.166 | 0.069 |
| Total fertility rate | -0.770 | < 0.001 | -0.749 | < 0.001 |
| Urbanization | 0.011 | 0.857 | 0.032 | 0.607 |

**Stepwise**

| Population ageing (65yo%) | | | | | |
| Nurse density (not added) | | | Nurse density (added) | | |
| Model | Variable | Adjusted $R^2$ | Model | Variable | Adjusted $R^2$ |
|---|---|---|---|---|---|
| 1 | Total fertility rate | 0.722 | 1 | Total fertility rate | 0.722 |
| 2 | Physician density | 0.729 | 2 | Nurse density | 0.732 |
| | GDP PPP | Insignificant | 3 | GDP PPP | 0.736 |
| | Obesity prevalence | Insignificant | 4 | Physician density | 0.741 |
| | Urbanization | Insignificant | | Obesity prevalence | Insignificant |
| | Nurse density | Not added | | Urbanization | Insignificant |

Significance level:

* p<0.05;

** p< 0.01;

***p< 0.001

Data source and definition Data source and definition: Nurse density, expressed with the number of nurses and midwives per 1,000 population (the Word Bank); Population ageing, presented with the percent of population aged 65 and above (the Word Bank); GDP PPP, the per capita purchasing power parity (PPP) value of all final goods and services produced within a territory in a given year (the Word Bank); Obesity prevalence, the percentage of population with BMI ≥30 prevalence (WHO Global Health Observatory); Physician availability, the number of nurses and midwives per 1,000 population (the Word Bank); Total fertility rate, representing the number of children that are born to a woman, the World Bank; Urbanization, the percentage of population living in urban area (the Word Bank); All the data were log-transformed for correlation analysis.

was "added" as a predicting variable, it showed its significant contribution to 65yo% together with total fertility rate (β = 0.178, p<0.010; β = -0.239, p<0.010; β = -0.749, p< 0.001, Table 3-1). Obesity prevalence, physician density and urbanization showed very weak and insignificant relationship to 65yo% (Table 3-1).

Similarly, in the subsequent stepwise linear regression model, when nurse density was "not added" as one of the predictors, total fertility rate and physician density were selected as the two most significant variables which totally explained 72.9% ($R^2$ = 0.729, Table 3-2) of 65yo% variance. While nurse density was "added" as a predicting variable, the stepwise linear regression model selected nurse density as the 2nd most influential predictor for 65yo% with ($R^2$ increment = 0.010, Table 3-2). In the stepwise regression analysis model, totally, 74.1% ($R^2$ = 0.741) of 65yo% variance was explained by the 4 selected significant predicting and potential confounding variables (total fertility rate, nurse density, GDPP PPP and physician density, Table 3-2).

Table 4 showed the relationship between nurse density and 65yo% in different country groupings. The best fit equations and R square extracted from the scatt plots consisting of

**Table 4. Bivariate correlations between nurse density and population ageing level within various country groupings.**

| Country grouping | Best fit equation | $R^2$ | Trendline pattern | Fisher's r-to-z transformation |
|---|---|---|---|---|
| UN common practice | | | | |
| Developed | $y = -0.4141x^2 + 1.7808x + 1.0889$ | $R^2 = 0.722$, n = 45 | Polynomial | Developed vs developing countries: |
| Developing | $y = 0.0167x^2 + 0.0891x + 0.6085$ | $R^2 = 0.248$, n = 141 | Polynomial | z = 4.02, p< 0.001 |
| World Bank income classifications | | | | |
| Low | $y = 0.0814x^2 - 0.6136x + 2.1834$ | $R^2 = 0.2908$ | Polynomial | |
| Low middle | $y = -0.0249x^2 + 0.5036x - 0.4412$ | $R^2 = 0.1924$ | Polynomial | |
| Upper middle | $y = 0.0381x^2 - 0.1774x + 1.7187$ | $R^2 = 0.1819$ | Polynomial | |
| High | $y = -0.1445x^2 + 2.5845x - 8.3844$ | $R^2 = 0.1590$ | Polynomial | |
| WHO Regions | | | | |
| AFRO | $y = 0.1026x^2 - 0.815x + 2.6237$ | $R^2 = 0.4013$ | Polynomial | |
| AMRO | $y = 0.0176x^2 + 0.0354x + 1.3525$ | $R^2 = 0.4841$ | Polynomial | |
| EMRO | $y = -0.167x^2 + 1.6365x - 2.4014$ | $R^2 = 0.3026$ | Polynomial | |
| EURO | $y = -0.1331x^2 + 0.8006x + 1.6858$ | $R^2 = 0.0552$ | Polynomial | |
| SEARO | $y = -0.1309x^2 + 1.4553x - 2.083$ | $R^2 = 0.1015$ | Polynomial | |
| WPRO | $y = 0.18x^2 - 1.634x + 5.0363$ | $R^2 = 0.6539$ | Polynomial | |
| Countries grouped based on various factors | | | | |
| ACD | $y = 0.6558ln(x) + 0.583$ | $R^2 = 0.0359$ | Logarithmic | |
| APEC | $y = 0.6242e^{0.2021x}$ | $R^2 = 0.6200$ | Exponential | |
| Arab World | $y = -0.143x^2 + 1.408x - 1.93$ | $R^2 = 0.2206$ | Polynomial | |
| EEA | $y = -0.2023x^2 + 0.957x + 1.8014$ | $R^2 = 0.3603$ | Polynomial | |
| EOL | $y = 0.1063x^2 - 0.7988x + 2.4714$ | $R^2 = 0.5424$ | Polynomial | |
| EU | $y = -0.1635x^2 + 0.6147x + 2.4294$ | $R^2 = 0.0776$ | Polynomial | |
| LA | $y = 0.0263x^2 - 0.0409x + 1.4964$ | $R^2 = 0.6468$ | Polynomial | |
| LAC | $y = 0.0031x^2 + 0.1594x + 1.1142$ | $R^2 = 0.4057$ | Polynomial | |
| OECD | $y = -0.2023x^2 + 0.957x + 1.8014$ | $R^2 = 0.3603$ | Polynomial | |
| SADC | $y = 0.1063x^2 - 0.7988x + 2.4714$ | $R^2 = 0.5424$ | Polynomial | |

Data source and definition: Nurse density, measured with the number of nurses and midwives per 1,000 population (the Word Bank); Population ageing, presented with the percent of population aged 65 and above (the Word Bank).

All the data were log-transformed for correlation analysis.

nurse density and 65yo% showed that nurse density universally correlated to 65yo% in all country groupings. The highlights of the results were the relationships between nurse density and 65yo% in United Nations developed and developing countries ($R^2 = 0.722$ and 0.248 respectively, Table 4). Fisher's r to z transformation revealed that nurse density had the significantly more important role in promoting 65yo% in developed countries than in the developing countries (z = 4.02, p< 0.001, Table 4).

## Discussion

This ecological study examined the correlation between nursing density and population ageing (65yo%) with consideration of the confounding effects of economic affluence (GDP PPP), obesity prevalence, physician density, total fertility rate and urbanization. The findings in this study reveal that nurse density may be a significant contributor to population ageing, and this contributing effect remains independent of economic affluence (GDP PPP), total fertility rate, obesity, physician density and urbanization.

It is well-known that, as the biggest healthcare professional group [10], nurses are essential to primary, secondary and tertiary health care. This is evident in the intrinsic correlation between nurse density and improvements to life expectancies at birth and 65 years of age, old within OECD countries [13] and worldwide [12].

The impact of nursing has been seen in many domains of health such as management of infectious diseases, a significant contributor to reduced life expectancy. As healthcare professionals in constant communication with patients and their needs nurses have led and manage infection control services across acute and community settings, effectively interrupting disease transmission on multiple levels of population and social status [47, 48]. As the frontline healthcare professional, nurses have been playing a crucial role in preventing COVID- 19 from spreading and treating COVID-19 patients [49, 50]. Nursing interventions have been reducing infectious disease mortality rates across the whole population, including the segment aged 65 and over [51].

High child mortality rates and the ongoing impact of an infectious disease pandemic are significant obstacles for people to survive to 65 years of age. Globally, nurses have been playing a pivotal role in reducing child mortality rate through heavy involvement in communicable and non-communicable disease management [11, 52, 53]. Thompson and Keeling report that over 40 years, improvements in child mortality have occurred through nurses educating and assisting young mothers on how to feed their children and how to maintain hygiene [52].

Patient education and public health promotion are critical components of the nurses' daily job, which may be even more important concerns for the older population because of the second epidemiology transition [54]. This transition, marked by a rapidly growing public health crisis concerning a shift from infectious diseases to the impact of chronic diseases is a consequence of good nursing care. This has been evidenced in the role of the nursing workforce in contributing to life expectancy both at younger age and then older than 65 across epidemics, acute health conditions [12, 13] and a range of degenerative diseases which occur in older populations [13]. In terms of approach, both studies also correlated nurse density to life expectancy at population level, similar to those revealed in this study.

Nurses are central to full coverage healthcare services, they are often the first and sometimes the only health professionals that patients see [10]. The quality of healthcare received from nurses has been associated with patient safety [55–60], low mortality rate [61–66] and patient outcomes [55, 59, 67–70]. While the breadth of the impact of nurses on people's health remains difficult to fully quantify, the community recognition of nursing healthcare services occurs through various awards, for example, the International Year of the Nurse and the Midwife [71] and the Gallup Roll [72].

## Strengths and limitations

The greatest strength of this study is free and unrestricted access to the predicting variable (nurse density) and dependent variable (65yo%) which was complimented with data concerning potential confounding variables (GDP PPP, obesity prevalence, total fertility rate and urbanization). The time based serial data enabled both predicting and confounding variables to observe their delayed presentations and impact on 65yo%. This enabled data analysis using several different models, such as scatterplots, bivariate, partial correlation and linear regression. This level of data availability and subsequent multiple data analysis approaches provide a rich source of findings when compared to individual based studies.

Some limitations to be noted:

Firstly, this is an ecological study, and the results are subject to ecological fallacy. The correlations identified in this study are at population level, but they may not necessarily hold true at the individual level.

Secondly, this is a cross sectional study, the relationship between nurse density and 65yo% is only correlational, not causal.

Thirdly, the data included in this study may be crude and have some random errors when the United Nations agencies collected and aggregated data at population level. However, the analysis results are highly repeatable which may be different from individual data based studies.

### Implications for practice

This study delivers a strong message to health authorities worldwide. As populations age 65 and over increase, the prevalence of disability, frailty and chronic diseases also rises [14, 73, 74]. With the increasing percentage of the world population aged 65+, the pressure for nursing care becomes cumulative, essentially exacerbating the challenges of a shortage of nursing staff. This study should encourage healthcare authorities, to consider the implications increase of density when promoting population health.

### Conclusions

Nurse density correlates to population ageing globally and regionally, but is more strongly in developed countries. This suggests that nurse density may be a significant predictor for population ageing, and is more prominent in developed countries. Ironically, nursing healthcare has been partially responsible for population ageing which in turn demands more nursing care and accordingly exposes the nursing profession to a worsening shortage of staff. To highlight the profound implications for the role the nursing profession plays especially at a time of global nursing shortage, further study into the effects of long-run elasticity of nurse staffing level on population ageing may be needed. For instance, what percentage of increase of nursing staff would be required to meet every 1% increase of an ageing population.

### Author Contributions

**Conceptualization:** Wenpeng You, Frank Donnelly.

**Data curation:** Wenpeng You.

**Formal analysis:** Wenpeng You, Frank Donnelly.

**Investigation:** Wenpeng You, Frank Donnelly.

**Methodology:** Wenpeng You, Frank Donnelly.

**Project administration:** Wenpeng You.

**Resources:** Wenpeng You, Frank Donnelly.

**Software:** Wenpeng You.

**Validation:** Frank Donnelly.

**Visualization:** Wenpeng You, Frank Donnelly.

**Writing – original draft:** Wenpeng You.

**Writing – review & editing:** Wenpeng You, Frank Donnelly.

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
