## [Decision Letter · Decision Letter 0]

27 Feb 2023

PONE-D-22-35617Greater nurse density correlates to higher level of population ageing globally, but is more prominent in developed countries.PLOS ONE

Dear Dr. You,

Thank you for submitting your manuscript to PLOS ONE. After careful consideration, we feel that it has merit but does not fully meet PLOS ONE’s publication criteria as it currently stands. Therefore, we invite you to submit a revised version of the manuscript that addresses the points raised during the review process.

We look forward to receiving your revised manuscript.

Kind regards,

Alice Mannocci, Ph.D, MS

Academic Editor

PLOS ONE

Journal Requirements:

Reviewers' comments:

Reviewer's Responses to Questions

**Comments to the Author**

1. Is the manuscript technically sound, and do the data support the conclusions?

Reviewer #1: Yes

Reviewer #2: Yes

Reviewer #3: Yes

2. Has the statistical analysis been performed appropriately and rigorously? 

Reviewer #1: Yes

Reviewer #2: Yes

Reviewer #3: Yes

3. Have the authors made all data underlying the findings in their manuscript fully available?

Reviewer #1: Yes

Reviewer #2: No

Reviewer #3: Yes

4. Is the manuscript presented in an intelligible fashion and written in standard English?

Reviewer #1: Yes

Reviewer #2: Yes

Reviewer #3: No

5. Review Comments to the Author

Reviewer #1: 1. The study presents the results of original research.

2. Results reported have not been published elsewhere.

3. Experiments, statistics, and other analyses are performed to a high technical standard and are described in sufficient detail.

4. Conclusions are presented in an appropriate fashion and are supported by the data.

5. The article is presented in an intelligible fashion and is written in standard English.

6. The research meets all applicable standards for the ethics of experimentation and research integrity.

7. The article adheres to appropriate reporting guidelines and community standards for data availability.

Reviewer #2: your Introduction part is shallow and needs great attention and modification, your method part needs grammer correction and you should devoite your time again on this part, the reason for your log transformation is not clear

Reviewer #3: After analyzing the article, the following changes need to be made.

1. Please check the grammar and writing of the whole text. If the background section，greater nurse density correlates to higher levels of population ageing for those aged 65 or over. Methods：Population level data was extracted for exploring the correlation between nurse density and population 。

2.The results section of the abstract .nursing density has the second greatest influence on population ageing. Can care density affect population aging? Or does the order of words need to be switched.

3. Introduction :We support this hypothesis through calculating the correlation between nursing healthcare level and prevalence of older people (percentage of people aged 65 or above). Is it inconsistent with what is expressed in the title?

4. The r and R of the result section Figure 2-2 need to be unified.

5. Total fertility rate and urbanization show very weak and insignificant correlations to population ageing level (Table 3). I assume it should be total fertility rate and Physician density rather than urbanization.

6. The discussion is too complicated, so we can go straight to the topic and analyze the result part in depth, and many contents of the text are not very relevant.

6. PLOS authors have the option to publish the peer review history of their article (what does this mean?). If published, this will include your full peer review and any attached files.

Reviewer #1: No

Reviewer #2: **Yes: **Worku Chekol Tassew

Reviewer #3: **Yes: **HuiTan

---

## [Author Response · Author response to Decision Letter 0]

14 Mar 2023

Please see our response to the reviewers' comments in the separate Word document, titled 0 Response to Review Decision. 

It has been uploaded.

---

## [Decision Letter · Decision Letter 1]

5 Apr 2023

PONE-D-22-35617R1

Greater nurse density correlates to higher level of population ageing globally, but is more prominent in developed countries.

PLOS ONE

Dear Dr. You,

Thank you for submitting your manuscript to PLOS ONE. After careful consideration, we feel that it has merit but does not fully meet PLOS ONE’s publication criteria as it currently stands. Therefore, we invite you to submit a revised version of the manuscript that addresses the points raised during the review process.

We look forward to receiving your revised manuscript.

Kind regards,

Alice Mannocci, Ph.D, MS

Academic Editor

PLOS ONE

Reviewers' comments:

Reviewer's Responses to Questions

**Comments to the Author**

1. If the authors have adequately addressed your comments raised in a previous round of review and you feel that this manuscript is now acceptable for publication, you may indicate that here to bypass the “Comments to the Author” section, enter your conflict of interest statement in the “Confidential to Editor” section, and submit your "Accept" recommendation.

Reviewer #2: All comments have been addressed

Reviewer #3: (No Response)

2. Is the manuscript technically sound, and do the data support the conclusions?

Reviewer #2: Yes

Reviewer #3: (No Response)

3. Has the statistical analysis been performed appropriately and rigorously? 

Reviewer #2: Yes

Reviewer #3: (No Response)

4. Have the authors made all data underlying the findings in their manuscript fully available?

Reviewer #2: Yes

Reviewer #3: (No Response)

5. Is the manuscript presented in an intelligible fashion and written in standard English?

Reviewer #2: Yes

Reviewer #3: (No Response)

6. Review Comments to the Author

Reviewer #2: You have tried to incorporate reviewer’s comments. It assure the ethical principles and follow publication ethics

Reviewer #3: The discussion section is not profound, can a brief sentence summarize this study?

The model can only be used as a reference. Is there any other literature to support care density effect population aging?

The main results of the conclusion section need to be written out

The conclusion drawn from relevant analysis can only be a hypothesis, but cannot accurately confirm a certain result. Data analysis is somewhat simple. Can we further conduct data analysis and explore at a deeper level.

7. PLOS authors have the option to publish the peer review history of their article (what does this mean?). If published, this will include your full peer review and any attached files.

Reviewer #2: **Yes: **worku chekol tassew

Reviewer #3: **Yes: **Hui Tan

---

## [Author Response · Author response to Decision Letter 1]

17 Apr 2023

We have updated a separate document for addressing the reviewer's comment.

---

## [Decision Letter · Decision Letter 2]

26 Jun 2023

PONE-D-22-35617R2Greater nurse density correlates to higher level of population ageing globally, but is more prominent in developed countries.PLOS ONE

Dear Dr. You,

Thank you for submitting your manuscript to PLOS ONE. After careful consideration, we feel that it has merit but does not fully meet PLOS ONE’s publication criteria as it currently stands. Therefore, we invite you to submit a revised version of the manuscript that addresses the points raised during the review process.

We look forward to receiving your revised manuscript.

Kind regards,

Alice Mannocci, Ph.D, MS

Academic Editor

PLOS ONE

Journal Requirements:

Additional Editor Comments (if provided):

I'm suggesting to the Authors in order to improve the quality of the manuscript to follow the comments of the review.

Reviewers' comments:

Reviewer's Responses to Questions

**Comments to the Author**

1. If the authors have adequately addressed your comments raised in a previous round of review and you feel that this manuscript is now acceptable for publication, you may indicate that here to bypass the “Comments to the Author” section, enter your conflict of interest statement in the “Confidential to Editor” section, and submit your "Accept" recommendation.

Reviewer #3: All comments have been addressed

2. Is the manuscript technically sound, and do the data support the conclusions?

Reviewer #3: Yes

3. Has the statistical analysis been performed appropriately and rigorously? 

Reviewer #3: Yes

4. Have the authors made all data underlying the findings in their manuscript fully available?

Reviewer #3: Yes

5. Is the manuscript presented in an intelligible fashion and written in standard English?

Reviewer #3: Yes

6. Review Comments to the Author

Reviewer #3: The author has made significant revisions to the suggestions mentioned above, and will further refine the language of the article to avoid ambiguity. Does the author have any methods, interventions, and subsequent research to address this topic? Does rationality and necessity exist, and is there any significance for further in-depth research. The author can briefly narrate

7. PLOS authors have the option to publish the peer review history of their article (what does this mean?). If published, this will include your full peer review and any attached files.

Reviewer #3: No

---

## [Author Response · Author response to Decision Letter 2]

7 Jul 2023

Please see attached document titled, 0 Response to Review Decision

---

## [Decision Letter · Decision Letter 3]

19 Sep 2023

Greater nurse density correlates to higher level of population ageing globally, but is more prominent in developed countries.

PONE-D-22-35617R3

Dear Dr. You,

We’re pleased to inform you that your manuscript has been judged scientifically suitable for publication and will be formally accepted for publication once it meets all outstanding technical requirements.

Kind regards,

Alice Mannocci, Ph.D, MS

Academic Editor

PLOS ONE
---

## [Editor Report · Acceptance letter]

22 Sep 2023

PONE-D-22-35617R3 

Greater nurse density correlates to higher level of population ageing globally, but is more prominent in developed countries. 

Dear Dr. You:

I'm pleased to inform you that your manuscript has been deemed suitable for publication in PLOS ONE. Congratulations! Your manuscript is now with our production department. 

Kind regards, 

on behalf of

Prof. Alice Mannocci 

Academic Editor

PLOS ONE